# Virtual Surgical Planning and Customized Subperiosteal Titanium Maxillary Implant (CSTMI) for Three Dimensional Reconstruction and Dental Implants of Maxillary Defects after Oncological Resection: Case Series

**DOI:** 10.3390/jcm11154594

**Published:** 2022-08-06

**Authors:** Jose Luís Cebrián Carretero, José Luis Del Castillo Pardo de Vera, Néstor Montesdeoca García, Pablo Garrido Martínez, Marta María Pampín Martínez, Iñigo Aragón Niño, Ignacio Navarro Cuéllar, Carlos Navarro Cuéllar

**Affiliations:** 1Oral and Maxillofacial Surgery Department, Hospital La Paz, Paseo de la Castellana, 261, 28046 Madrid, Spain; 2Oral and Maxillofacial Surgery Department, Hospital La Luz, 28003 Madrid, Spain; 3Oral and Maxillofacial Surgery Department, General Universitario HLA Moncloa, Avenida de Valladolid, 83, 28008 Madrid, Spain

**Keywords:** subperiosteal maxillary implants, 3D reconstruction, oral rehabilitation

## Abstract

Maxillectomies cause malocclusion, masticatory disorders, swallowing disorders and poor nasolabial projection, with consequent esthetic and functional sequelae. Reconstruction can be achieved with conventional approaches, such as closure of the maxillary defect by microvascular free flap surgery or prosthetic obturation. Four patients with segmental maxillary defects that had been reconstructed with customized subperiosteal titanium maxillary implants (CSTMI) through virtual surgical planning (VSP), STL models and CAD/CAM titanium mesh were included. The smallest maxillary defect was 4.1 cm and the largest defect was 9.6 cm, with an average of 7.1 cm. The reconstructed maxillary vertical dimension ranged from 9.3 mm to 17.4 mm, with a mean of 13.17 mm. The transverse dimension of the maxilla at the crestal level was attempted to be reconstructed based on the pre-excision CT scan, and these measurements ranged from 6.5 mm in the premaxilla area to 14.6 mm at the posterior level. All patients were rehabilitated with a fixed prosthesis on subperiosteal implants with good esthetic and functional results. In conclusion, we believe that customized subperiosteal titanium maxillary implants (CSTMI) are a safe alternative for maxillary defects reconstruction, allowing for simultaneous dental rehabilitation while restoring midface projection. Nonetheless, prospective and randomized trials are required with long-term follow-up, to assess its long-term performance and safety.

## 1. Introduction

The maxilla is a very important structure in the osseous structure of the face. Large maxillary defects secondary to trauma, congenital malformations and tumor resections cause serious bone and soft tissue defects, with consequent esthetic and functional sequelae [1]. Maxillectomy causes malocclusion, disorders of mastication, swallowing, speech and a deficient nasolabial projection and, consequently too of the midfacial center [2]. Its reconstruction requires independence of the oral, nasal and orbital cavities to achieve a functional reconstruction and bone restoration, as well as soft tissue support, which is a real challenge for the reconstructive surgeon [3,4].

Among the options employed during the last decades, the use of conventional techniques such as regional flaps (fascia and temporal muscle), prosthetic obturators, prostheses supported on zygomatic implants and the use of microvascularized flaps stand out [5,6]. However, the sealing of the maxillary defect by means of soft parts does not solve the problem of masticatory function, the use of prosthetic obturators [7] causes social and psychological problems for the patient that alter his quality of life and, sometimes, the use of zygomatic implants does not provide sufficient anchorage and stability in the prostheses [8].

For this reason, in recent years, reconstruction using microvascularized free flaps has become the technique of choice, since it allows for immediate reconstruction with a fairly predictable result and sometimes allows the placement of osseointegrated dental implants. The ultimate goal of reconstruction is to allow optimal dental rehabilitation [1,2,3].

The functions of the oral cavity depend on several factors, these being the volumetric reconstruction of the maxillary defect and the type of tissue used in the reconstruction (taking into account that sometimes if postoperative radiotherapy is required, the tissues lose stability and harden, which can hinder the placement and retention of a prosthesis), so that the placement of implants in the bone flaps are increasingly used to achieve adequate oral rehabilitation [5,9].

The most commonly used vascularized free flaps are the fibula flap, the antebrachial flap, iliac crest flap, rectus abdominis muscle flap, inferior abdominal artery perforator flap and the anterolateral thigh perforator flap. Among those that provide bone that could allow the placement of dental implants, they sometimes do not provide sufficient bone height to restore the maxillary ridge [10]. This discrepancy, as occurs in mandibular reconstruction, causes great difficulty in implant placement, causing overloading of the implants, with the risk of loss of the functional and esthetic result in the medium/long term. Therefore, a reconstruction is required that allows an adequate volumetric restitution for the optimal function of the implant-supported structure.

Subperiosteal implants were developed in Sweden at the beginning of the 1940s. Positioning these implants in the patient was very difficult and could cause a range of complications. In recent years, the digital revolution has meant a paradigm shift in the world of oral and maxillofacial surgery.

The purpose of this study was to evaluate the outcomes of the three-dimensional reconstruction of segmental maxillary defects with customized subperiosteal titanium maxillary implants (CSTMI) through virtual surgical planning (VSP), STL models and CAD/CAM titanium mesh [11]. STLs (stereolithographic models) are the files that include the three-dimensional information of the anatomical models; CAD/CAM (Computer-Aided Design and Manufacturing) is the process of three-dimensional design and fabrication of titanium mesh in a patient-specific manner through computer design and additive manufacturing via 3D printing.

## 2. Materials and Methods

### 2.1. Patients

To address the research purpose, the investigators performed a retrospective study, including patients with segmental maxillary oncological defects that had been reconstructed with a subperiosteal titanium maxillary implant at Hospital General Universitario La Paz (Madrid, Spain) between 2018 and 2021. Four oncologic patients were diagnosed with maxillary squamous cell carcinoma (Table 1).

Due to the difficulty or impossibility of bone reconstruction of the maxilla with the conventional techniques of bone grafting, microsurgical reconstruction or osteogenic distraction, all patients had previously used a removable dental prosthesis.

### 2.2. Preoperative Planning

Virtual surgical planning (VSP), stereolithographic models (STL) and a custom-made titanium meshes (CAD/CAM) (Avinent®, Madrid, Spain) were designed prior to surgery to enable both vertical and horizontal reconstruction of the maxillary defect. The surgery was planned with the help of high-resolution computed tomography using 0.5 mm thin slices, and plaster models were used to plan the optimal position of the dental crowns.

The DICOM files obtained from the CBCT were imported into software to create three-dimensional (3D) models of the residual bone anatomy in each patient. Each file obtained was saved in STL format and then merged with the STL file obtained from the intraoral scanner and the diagnostic wax-up to improve all the patient’s bone and dental information and achieve greater precision in the final result. This step made it possible to determine the ideal prosthetic emergence profile and to plan the position of the implants appropriately.

The next step was to design and define the shape and extent of the subperiosteal structure, taking into account the position of the prosthetic abutments and the remaining bone in each case. For this reason, the areas of greatest thickness and bone density were chosen for the location of the screws that would fix the structure, as well as the length of each screw in its specific position.

The final structure was exported in STL format to review and polish the edges and maximize the 3D quality of the design, making it ready for manufacturing.

### 2.3. Surgical Procedure

All procedures were performed under general anesthesia. A single injection of 2 g of amoxiclavulanic acid was administered intraoperatively. After infiltration with a local anesthetic and vasoconstrictor (articaine and 1:200.000 epinephrine), crestal incisions were performed to raise a mucoperiosteal flap to ensure adequate soft tissue coverage of the titanium implant.

Then, strict subperiosteal dissection of the alveolar ridge of the defect to be reconstructed and the adjacent areas including the remaining teeth was performed. After careful dissection with a periostotome and complete exposure of the maxillary defect, the maxillary implant was placed, ensuring a passive fit of the CSTMI on the bone surface. The CSTMI was fixed to the bone using 1.5 and 2 mm diameter screws of different lengths according to the previous virtual plan. The ends of the prosthetic connections emerged through small incisions in the flap. Finally, periosteal relaxing incisions were made in the periosteum to favor a correct mobilization of the flap, and the wound was sutured in a watertight fashion with resorbable sutures.

In the postoperative period, oral antibiotic therapy (amoxicillin/clavulanic acid 1 g/8 h) was prescribed for the first 7 days, along with analgesics, anti-inflammatory drugs and 0.12% chlorhexidine mouthwashes, two or three times a day, during the first week.

### 2.4. Prosthodontic Rehabilitation

Two weeks after the intervention, the stitches are removed and a provisional prosthesis is placed by the prosthodontist, milling the edges to avoid friction and mucosal injuries.

A month and a half after the operation, once the gum is completely healed, impressions are taken, and the fixed prosthesis is made in metal porcelain designed by CAD/CAM. All the prostheses were made in the clinic using the conventional protocol.

### 2.5. Follow-Up Visits

All patients were followed up periodically to detect any complications regarding soft tissue coverage, prosthetic complications or peri-implantitis or hardware failure (Table 1).

Esthetic results: An esthetic evaluation was performed by the patients to assess scores in facial symmetry, intraoral healing and maxillary projection. The results were classified with scores 0 (“poor”), 1 (“fair”) and 2 (“good”).

Functional results: All patients were rehabilitated with implant-supported fixed prostheses. Swallowing was evaluated and the results were classified with the following scores:

0 (liquid diet), 1 (soft diet) and 2 (normal diet). Speech articulation was assessed as intelligible speech and unintelligible speech.

## 3. Results

Four patients with segmental maxillary defects that had been reconstructed with customized subperiosteal titanium maxillary implants (CSTMI) through virtual surgical planning (VSP), STL models and CAD/CAM titanium mesh were included. 

Due to the difficulty or impossibility of bone reconstruction of the maxilla with the conventional techniques of bone grafting, microsurgical reconstruction or osteogenic distraction, all patients had previously used a removable dental prosthesis.

The follow-up period was from 9 months to 3 years 2 months (averaging 1 year and 8 months). In all patients, the diagnosis was squamous cell carcinoma of the maxilla.

Microvascularized free graft reconstruction was attempted in three patients, but failed. In the remaining patient, a maxillary defect was reconstructed after oncological excision, as their general condition made microsurgical reconstruction inadvisable. The patients had an average age of 66.2 years. Three patients were men (75%), and one was a woman (25%). The smallest maxillary defect was 4.1 cm and the largest defect was 9.6 cm, with an average of 7.1 cm (Table 1).

Two patients received previous radiotherapeutic treatment (50%), and subperiosteal implant placement was delayed up to 2 years in these radiated cases and was only one year after excision in the two non-irradiated cases. The reconstructed maxillary vertical dimension ranged from 9.3 mm to 17.4 mm, with a mean of 13.17 mm. The transverse dimension of the maxilla at the crestal level was attempted to be reconstructed based on the pre-excision CT scan, and these measurements ranged from 6.5 mm in the premaxilla area to 14.6 mm at the posterior level.

All patients evolved uneventfully, and there were no infection signs or soft tissue dehiscence, and the oral mucosa healed perfectly without recessions or ulcers. The immediate postoperative period was painless in all patients, with only slight discomfort.

All implants were perfectly adjusted to the planned position, as confirmed by postoperative CBCT. At the end of the follow-up period, none of the patients had pain or soft tissue or prosthetic part problems. The biocompatibility of the patients with the materials used was good, and no complications were found in the soft tissues, the maxillary bone, the implants or the dental prostheses (Table 1).

No exposure of the titanium mesh was observed, and in none of the patients was the particulate bone graft used. All patients were rehabilitated with a fixed implant prosthesis.

All patients reported a good esthetic result. In terms of functional results, speech articulation was evaluated as intelligible language in all patients. All patients reported a regular diet.

## 4. Case Presentation

A 45-year-old male patient came to our department with mobility of teeth in the premaxilla and associated palatal lesion (Figure 1A,B). After histological and radiological studies, a diagnosis of maxillary squamous cell carcinoma was made. Tumor resection with segmental maxillectomy and clear margins and immediate reconstruction with a three-segment fibula flap was performed (Figure 1C,D).

He underwent surgery two years later due to contralateral recurrence and was treated with another fibula flap and radiotherapy in another institution (Figure 2A). He came to our department because he had problems with implants, which were subsequently removed. (Figure 2B,C).

Two and a half years after the beginning of the treatment, a volumetric reconstruction and dental rehabilitation of the maxillary defect by means of a custom-made subperiosteal titanium maxillary implant were proposed.

The DICOM data obtained from the CBCT were extracted and imported into software, where the residual anatomy of the patient’s bone was reconstructed in 3D (Mimics^®^, Materialise, Leuven, Belgium), and the file was saved as an STL. In this phase, care was taken to best define the cortical walls of the residual bone (Figure 3A,C). The best position for the fixation screw was also evaluated. 

Virtual planning (Materialise Mimics v22.0^®^, Materialise Iberia NV, Barcelona, Spain) was performed for placement of a customized prosthesis. The 3D reconstruction was then aligned with the STL files obtained from the intraoral scan of the patient´s arch, and with the diagnostic wax-up. This helped define the optimal prosthetic emergence profile and allowed a proper design of the subperiosteal implant (Figure 4).

The implant was manufactured by Avinent^®^, Avinent Implant System, Barcelona, Spain, in sintered titanium with four subperiosteal implants with a universal external connection of 4.1 mm width. In addition to the structure, we used a 3D model of the patient’s maxilla and a replica of the structure to serve as a reference for the surgery (Figure 5).

The surgery was performed under general anesthesia and nasotracheal intubation. After local infiltration with a vasoconstrictor, a crestal incision was made extending towards posterior sectors, and a careful detachment of the periosteum, both in the vestibule and in the palatal area, allowed the placement of the custom-made structure without interference. The subperiosteal implant was placed in the planned area to check its correct adaptation to the bone, without forcing its positioning, and finally it was fixed to the remaining nasomaxillary bone and zygomaticomaxillary buttress by means of osteosynthesis screws predetermined in length according to the bone thickness in each area (Figure 6).

Two weeks after surgery, under local anesthesia, the attached gingiva was incised on each of the implant abutments, which were covered by the gingiva, and by means of straight and 30º pilates, a provisional prosthesis was made, which the patient used during the first 2 months. After this period, impressions were taken with an open tray for the definitive prosthesis. A metal framework was built in CAD/CAM milled cobalt–chrome and covered with feldspar ceramic (Figure 7A,C).

## 5. Discussion

Dental rehabilitation after oncological surgical treatment is of utmost importance and should be planned from the beginning [12,13]. Generally, endosseous implants can reliably be placed in fibula flap (FFF) primarily (immediately at the time of fibula harvest) or secondarily (delayed by 6–12 months), with comparable safety and outcome profile [14]. The most common complications associated are peri-implantitis and marginal bone loss. Nonetheless, the main drawback of the FFF bone is the lack of vertical height for implant placement

Another option is zygoma implants, combining autologous soft tissue reconstruction with zygomatic implant-supported rehabilitation. They provide a predictable support for prosthetic rehabilitation of the maxilla and can be placed at the time of ablative surgery [12]. However, occasionally, zygoma implants cannot be placed due to insufficient bone support, requiring excellent surgical skills. In addition, complications such as infection at the implant tip, tissue retraction, communication between the oral cavity and the maxillary sinus, extraoral fistulation and intra-orbital abscesses can occur [15].

On the other hand, microsurgical reconstruction of maxillary defects is complex and entails a high risk of [15,16]. Some patients may not be fit for microsurgical reconstruction due to poor basal status. In such patients, an alternative to accomplish adequate oral rehabilitation is to design patient-specific subperiosteal implants. 

In recent years, the advent of titanium 3D printing and 3D planning software have made reconsideration of the subperiosteal implant concept possible. The design of a subperiosteal implant is performed with a high degree of accuracy for each patient [17]. Thus, the concept of a ‘high-tech’ subperiosteal implant has been reborn.

Recently, Mommaerts introduced an innovative device for an additively manufactured subperiosteal implant (AMSJI®, CADskills, Gent, Belgium) that uses modern computer-aided design and manufacturing (CAD/CAM) technology [18] to provide an alternative implant option for dental rehabilitation in patients with extreme jawbone atrophy, which could be used for the rehabilitation of extended post-resection defects [13]. The major advantage of the subperiosteal implant is that it allows for immediate masticatory, esthetic and phonetic function [18].

Several papers have reported on the stability and osseointegration of the AMSJI at the wings and basal frame. In the study by Van den Borre et al., 15 patients who received bilateral AMSJI implantation in the maxilla were analyzed. Minor atrophy was seen at the alveolar ridge, but minimal atrophy was detected under the fixation wing. Patients showed a mean resorption at the alveolar ridge of 0.33 mm (SD 0.76 mm) and 0.08 (SD 0.33) mm at the wings and basal frame on the underlying zygo-maxillary bone one year post loading [17].

In a retrospective clinical study, Cerea et al. presented an analogue–digital technique for fabricating custom-made subperiosteal implants for both the maxilla and the mandible. In total, 70 partially or completely edentulous patients were included. At the two-year follow-up, the implant survival rate was 95.8%. The rate of prosthetic complications amounted to 8.9%. The authors concluded that the application of custom-made titanium subperiosteal implants can represent a safe alternative to conventional bone regeneration [19].

De Moor et al. [15] used finite element analysis to perform a biomechanical evaluation of the subperiosteal implant for the maxilla in a Cadwood and Howell class V patient. The simulations showed a stable and safe performance during average occlusal forces. However, they stated that further resorption of the residual ridge might cause fatigue of the implant, and thus, they recommend a close follow-up of bone quality in these patients. Additionally, the results showed that the arms experienced higher stresses compared to the rest of the implant, so future optimization should be driven towards the strengthening of the arms to improve stability. They concluded that this type of device is completely safe to use and is considered the best solution for edentulous patients with Cadwood and Howell class V–VIII bone resorption [15]. These results could be extrapolated and applied in cases of maxillary reconstruction in oncological patients.

The subperiosteal implant is a valuable and interesting alternative to major microsurgical reconstruction requiring composite bone free flaps, to bone grafting and to zygoma implants. Mastication can be provided immediately with one surgical intervention, and donor site morbidity is diminished, reducing complications and costs. With the new advances in virtual surgical planning and CAD/CAM technologies, specific subperiosteal implants with an optimal and accurate design can be manufactured.

The current digital fabrication procedure reduces the number of surgical sessions and patients’ discomfort. Additionally, customization makes the fit of the implant very accurate, which might ultimately reduce the number of intraoperative complications, increase the safety of the surgical procedure and improve the confidence of the surgeon. 

The ability to keep the fixation screws outside the maxillary sinus is also another major benefit of the subperiosteal implant in patients with chronic or silent sinusitis [18,19]. There is also the possibility to disconnect each post from the basal structure using a diamond burr under local anesthesia in cases of peri-implantitis without jeopardizing masticatory function, all while retaining the same prosthesis [19,20].

However, its efficiency still needs to be proven in the long term. Several reviews have, however, also reported on complications such as infections, early and late implant exposure, bone resorption, fistulation and implant mobility, leading to implant failure [17]. Prospective and more extensive studies are needed to assess its efficiency and safety and draw more conclusions. One new modification currently being evaluated by Mommaerts involves placing all the connecting arms under the palatal gingiva [18].

The main complications and problems could be related to material fatigue, poor osseointegration, implant exposure and mobility, peri-implantitis and length of the connection pillars used [21,22]. Randomized and prospective clinical trials are needed to assess the long-term performance of these devices. 

## 6. Conclusions

In conclusion, we believe that customized subperiosteal titanium maxillary implants (CSTMI) could be a safe alternative for maxillary defect reconstruction, allowing simultaneous dental rehabilitation while restoring midface projection. Recent reports and our experience show promising results in terms of functional and esthetic restoration. Nonetheless, prospective and randomized trials are required with long-term follow-up to assess its long-term performance and safety. 

We have reviewed the article against The PROCESS 2020 Guideline: Updating Consensus Preferred Reporting Of CasESeries in Surgery (PROCESS) Guidelines [23].

## Figures and Tables

**Figure 1 jcm-11-04594-f001:**
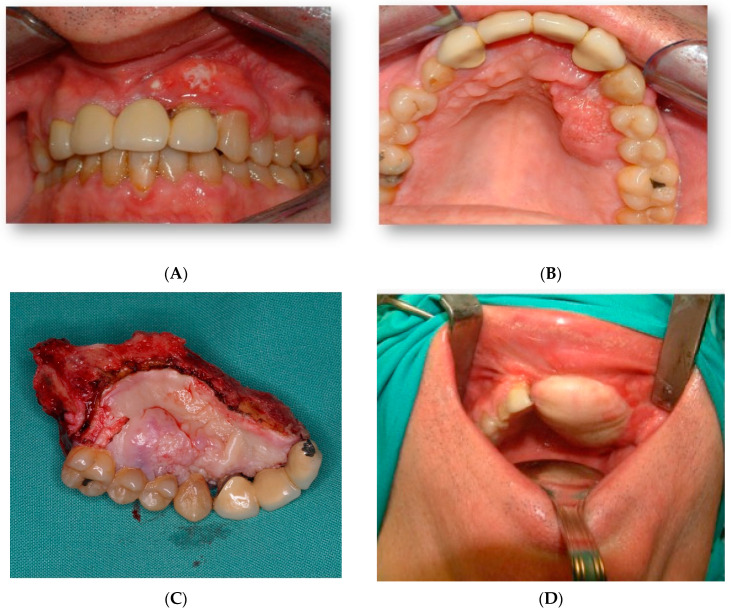
Intraoral view. (**A**,**B**) Maxillary squamous cell carcinoma with mobility of teeth. (**C**) Resection piece after excision of the tumour. (**D**) Microvascularized fibula flap.

**Figure 2 jcm-11-04594-f002:**
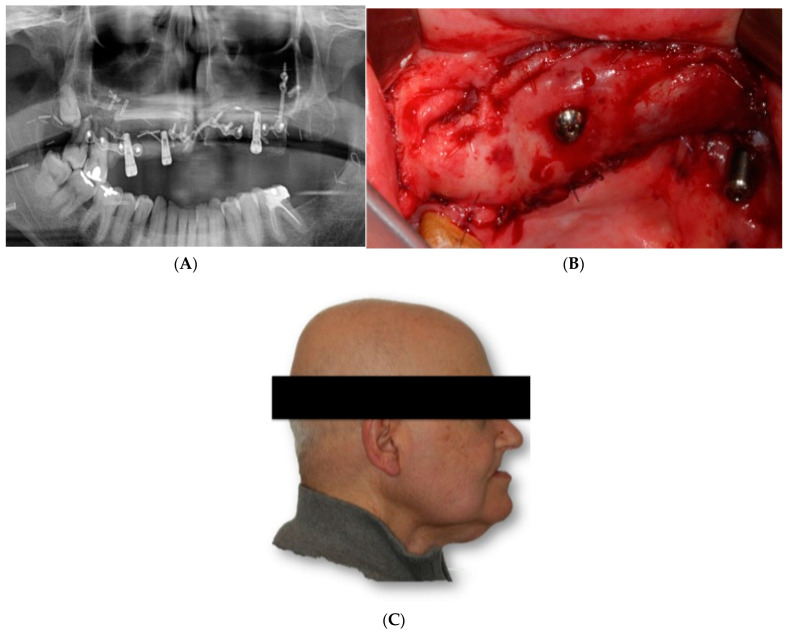
Bilateral fibula flap. (**A**) Ortopantomography showing both fibula flaps and the dental implants. (**B**) Intraoral view. (**C**) Clinical picture. Lack of maxillary projection.

**Figure 3 jcm-11-04594-f003:**
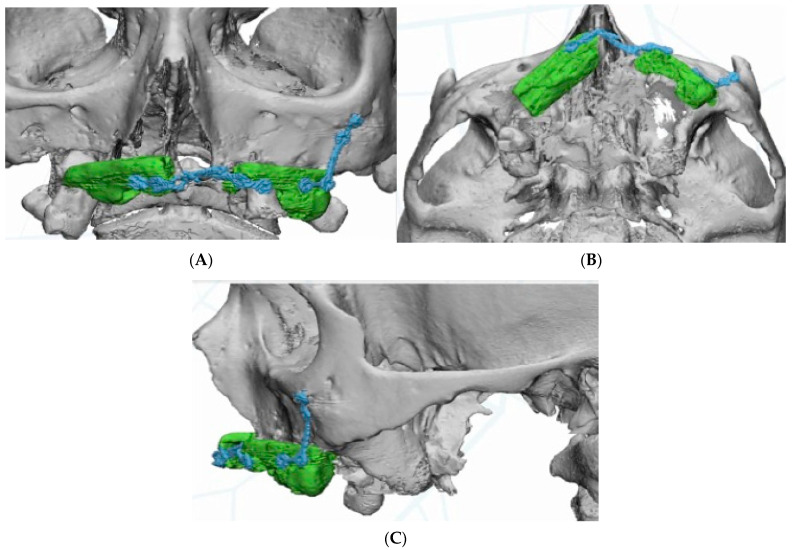
3D reconstruction of the residual bone of the fibula flaps. (**A**) Frontal view. (**B**). Basal view. (**C**) Lateral view.

**Figure 4 jcm-11-04594-f004:**
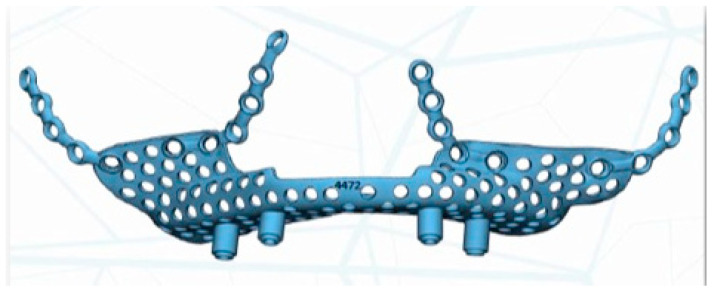
A 3D proper design of the subperiosteal implant.

**Figure 5 jcm-11-04594-f005:**
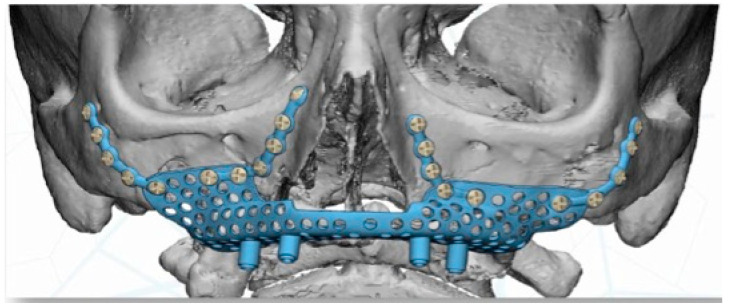
Virtual planning for placement of a customised prosthesis.

**Figure 6 jcm-11-04594-f006:**
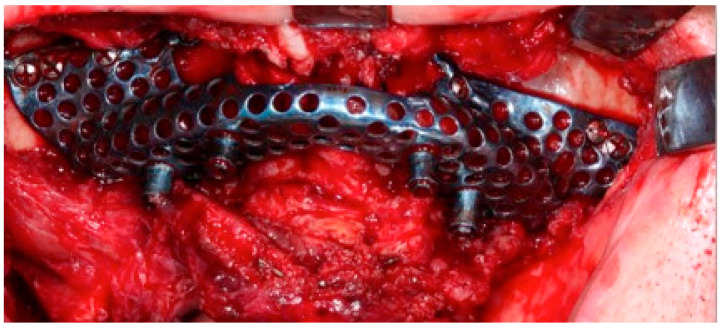
The customised prosthesis was positioned, which was fixed into the nasomaxillary and zygomaticomaxillary buttresses.

**Figure 7 jcm-11-04594-f007:**
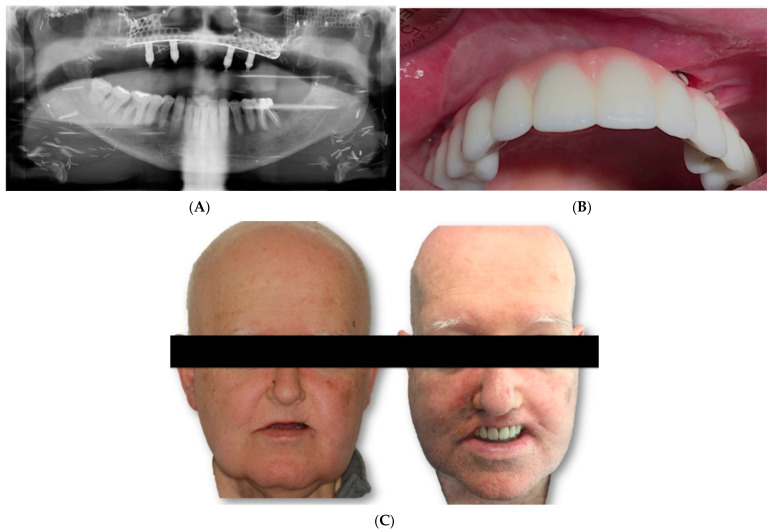
3D reconstruction of the residual bone of the fibula flaps. (**A**) Ortopantomography showing the subperiostal implant. (**B**) Definitive prostheses. (**C**) Before and after of the definitive prosthesis.

**Table 1 jcm-11-04594-t001:** Descriptive variables in all patients.

Gender Age (Years)	Diagnosis	Lenght of Defect (cm)	Vertical Reconstruction (mm)	Number of Implants	Radiotherapy	Functional Result	Aesthetic Result
M/59	Maxillary Squamous Cell Carcinoma	8.4	15.8	4	Yes	2	2
M/69	Maxillary Squamous Cell Carcinoma	6.3	10.2	6	Yes	2	2
F/65	Maxillary Squamous Cell Carcinoma	4.1	9.3	4	No	2	2
M/72	Maxillary Squamous Cell Carcinoma	9.6	17.4	6	No	2	2
Average		7.1	13.17				

## Data Availability

The data presented in this study are available on request from the corresponding author. The data are not publicly available due to data protection regulations.

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
