# Peer review of "Virtual Surgical Planning and Customized Subperiosteal Titanium Maxillary Implant (CSTMI) for Three Dimensional Reconstruction and Dental Implants of Maxillary Defects after Oncological Resection: Case Series"

_jcm, 2022, doi:10.3390/jcm11154594_

Round 1

Reviewer 1 Report

The paper titled “Virtual surgical planning and customized subperiosteal titanium maxillary implant (CSTMI) for three dimensional reconstruction and dental implants of maxillary defects after oncological resection” is interesting, which elaborated conducting to evaluate the outcomes of the three-dimensional reconstruction of segmental maxillary defects with customized subperiosteal titanium maxillary implants (CSTMI) through virtual surgical planning (VSP), STL models and CAD/CAM titanium mesh. The authors conclude that customized subperiosteal titanium maxillary implants (CSTMI) are a safe alternative for maxillary defects reconstruction, allowing simultaneous dental rehabilitation while restoring midface projection.

Comments

-         In title, it should be described the design of this study as Case Series.

-         How long these patients were followed-up? As well known, unfortunately, the long-term survival rate of subperiosteal dental implant is low as 75-79% and 50-60% in 10 and 15 years, respectively.

[Goldberg NI (1980) Risk of subperiostal implants. In: Proceedings of an NIH-Harvard development conference on dental implants-benefit and risk. U.S. Department of Health and Human Services, Public Health Service, National Institutes of Health, Bethesda, MD 89–94] [James RA, Lozada JL, Truitt PH, Foust BE, Jovanovic SA (1988) Subperiosteal implants. J Calif Dent Assoc 16:10–14] [Schou S, Pallesen L, Hjörtung-Hansen E, Pederson C, Fibäk B (2000) A 41-year history of a mandibular subperiostal implant. Clin Oral Impl Res 11:171–178]

-         With regarding the poor long-term outcome, osseointegrated zygoma implants have been suggested as the authors mentioned in discussion. The authors stated that the zygomatic implants needs for excellent surgical skills considering the maxillary sinus. However, it could be much easier through virtual surgical planning (VSP).

-         The disadvantage of Subperiosteal Implants and The Authors’ Special Solution should be clearly presented in discussion section to prevent typical complications of subperiosteal dental implant such as implant exposure, inflammation, infection, fistula formation, and severe atrophy of jaw bone during the function.

-         The discussion should be shortened to about 50%, and condensed on to improve the long-term outcome and to emphasize advantages over the zygomatic implant.

In summary,  I recommend first to address the above-mentioned issues. Thank you once again for submitting this interesting research.

Author Response

Thank for your comments.

We have included in the title "Case series".

The follow-up period was from 9 months to 3 years 2 months (average 1 year
and 8 months).

These articles are old, before the subperiosteal implants did not have the qualities of the material of today and they broke and infected much more, that's why they were abandoned by the endoosseous implants.
The current survival rate is much better.

Indeed, the use of mandibular subperiosteal implants is questionable due to the
high rate of local complications. But in the case of the maxilla, the fact that its
anchorage is located in the nasomaxillary and pyramidal buttresses, which
gives it extra support and, being far from the emergency mucosal areas of the
implant abutments, minimizes the occurrence of complications such as
infection, exposure of the titanium material, fistulas...

We have shortened the discussion and explained the advantages over the zygomatic implants.

Reviewer 2 Report

The manuscript s good and interesting.

It would be advisable to prepare a table of the previous retrospective clinical studies of subperiostal maxillary implants featuring the following items : diagnosis,defect sise,result and the complications.

Author Response

Thank you for the interesting suggestion. Nonetheless we believe a table of previous retrospective clinical studies would not contribute specially, since most reports (Vosselman et al. 2018, PMID: 30041913; Mommaerts et al. 2017 PMID: 28258795; Van den Borre et al. 2022 PMID: 34074574) are single case reports with a short-term follow up, bringing out the need for large case series with long term follow up. Most of these articles do not report defect size (except that most of them present maxillary defects) nor any remarkable complication. We believe in the future, a thorough review would be interesting when bigger patient size articles are available.

Reviewer 3 Report

Dear Authors,

The manuscript is a representation of four cases to evaluate the virtual surgical planning and customized subperiosteal titanium maxillary implant (CSTMI) for three dimensional reconstruction and dental implants of maxillary defects after oncological resection. I would suggest that the manuscript should be modified as per the comments .

Introduction

1.       Cite references for line no 71-74

2.       The authors wants to evaluate the outcomes of the three dimensional reconstruction of segmental maxillary defects with customized subperiosteal titanium maxillary implants (CSTMI) through virtual surgical planning (VSP), STL models and CAD/CAM titanium mesh. However, the introduction does not explain what is CSTMI. Neither it explains what are STL models and CAD/ Cam titanium mesh. It is suggested to dedicate one paragraph for the same.

3.       The introduction lacks the previous study done on the same topic . kindly add the research gap .

Discussion :

The discussion is too long. The initial few paragraphs were repetition what has been discussed in introduction. I would suggest omitting those paragraphs.

Conclusion : It should be modified as only four cases were assessed . Thus, conclusion should be suggestive rather than affirmative. 

Author Response

Thank for your comments.

1. References for line 71-74:

  1. Lenox ND, Kim DD. Maxillary reconstruction. Oral Maxillofac Surg Clin North Am. 2013 May;25(2):215-22.
  2. Chou PY, Denadai R, Hallac RR, Dumrongwongsiri S, Hsieh WC, Pai BC, Lo LJ. Comparative Volume Analysis of Alveolar Defects by 3D Simulation. J Clin Med. 2019 Sep 6;8(9):1401.

2. We have included explanations of the terms STL, CAD-Cam, CSTMI.

3. We have included in the introduction an outline of the studies on the subject of the article.

4. We have shortened the discussion by avoiding repetition in the introduction.

5. The conclusion suggests, but does not affirm the use of subperiosteal implants in maxillary defects:

In conclusion, we believe that customized subperiosteal titanium maxillary implants (CSTMI) could be a safe alternative for maxillary defects.

Round 2

Reviewer 1 Report

Much improved manuscript. However, I still have two concerns.

The authors answered that maxillary subperiosteal implant has a good prognosis because it is different from that of the mandible. I agree with your comments, but I still think that there should be a more sufficient rationale or Author's opinion to compensate for the disconnection of interest in the subperiosteal implant, which has been known as poor long-term outcome.

Some recent studies still have concerns about the accuracy of CT guided surgery. [Ku, JK., Lee, J., Lee, HJ. et al. Accuracy of dental implant placement with computer-guided surgery: a retrospective cohort study. BMC Oral Health 22, 8 (2022).] Please address the error and tolerances of the 3D method in this study.

Author Response

Thank for your comments.

We performed an accuracy study in orthognatic surgery evaluating the overall predictability of virtual surgical planning (Pampin Martinez MM et al. Evaluation of the Predictability and Accuracy of Orthognathic Surgery in the Era of Virtual Surgical Planning. Appl. Sci. 2022, 12(9), 4305. DOI: https://doi.org/10.3390/app12094305) and found that virtual planning and 3D printed wafers allow precise execution of the surgical planning, finding an error below 1mm for all studied cephalometric points. Blumer et al. reviewed patients who underwent orbital reconstruction using patient specific implants and found a mean surgical error of 0.6mm (Blumer et al. Surgical Outcomes of Orbital Fracture Reconstruction Using Patient-Specific Implants. J Oral Maxillofac Surg. 2021 Jun;79(6):1302-1312 DOI: 10.1016/j.joms.2020.12.029).

Ruckschlob et al. analyzed the accuracy of patient-specific implants and a3D printing of surgical splints in orthognathic surgery and found that thr PSI group showed an overall higher accuracy compared to the 3D printed splint one, especially for anterior/posterior translational mo9¡ovement (p<0.002) (Rückschlob et al. Accuracy of patient-specific implants and additive-manufactured surgical splints in orthognathic surgery - A three-dimensional retrospective study. J Craniomaxillofac Surg 019 Jun;47(6):847-853 DOI: 0.1016/j.jcms.2019.02.011). Alqussair et al. also evaluated the surgical accuracy of positioning the maxilla in orthognatic surgery through a CAD/CAM approach. They observed that mean absolute deviations were 0.98, 0.67, and 0.62 mm in the sagittal, vertical, and transverse coordinates, respectively. (Alqussair et al. Surgical Accuracy of Positioning the Maxilla in Patients With Skeletal Class II Malocclusion Using Computer-Aided Design and Computer-Aided Manufacturing-Assisted Orthognathic Surgery. J Craniofac Surg. 2021 Dec 13 DOI: 10.1097/SCS.0000000000008407). Moellmann et al. analyzed the accuracy of PSI implants in craniofacial reconstruction. They stablished a tolerance threshold of 1.5mm and found that on average, 95.17% (SD = 9.42) of the measurements between planned and surgically achieved implant position were within the defined tolerance range (Moellmann et al. Evaluation of the Fitting Accuracy of CAD/CAM-Manufactured Patient-Specific Implants for the Reconstruction of Cranial Defects-A Retrospective StudyJ Clin Med. 2022 Apr 6;11(7):2045 DOI: 10.3390/jcm11072045)

Therefore, we believe that a level of tolerance below 1mm should be considered for PSI reconstruction. In line with are previous article, we are currently carrying out a study to evaluate the accuracy of PSI implants and surgical guides in orthognathic surgery and preliminary data show minimal error (below 1mm).